# EvalNE: A Framework for Evaluating Network Embeddings on Link Prediction

## Abstract

Network embedding (NE) methods aim to learn low-dimensional representations of network nodes as vectors, typically in Euclidean space. These representations are then used for a variety of downstream prediction tasks. Link prediction is one of the most popular choices for assessing the performance of NE methods. However, the complexity of link prediction requires a carefully designed evaluation pipeline to provide consistent, reproducible and comparable results. We argue this has not been considered sufficiently in recent works. The main goal of this paper is to overcome difficulties associated with evaluation pipelines and reproducibility of results. We introduce EvalNE, an evaluation framework to transparently assess and compare the performance of NE methods on link prediction. EvalNE provides automation and abstraction for tasks such as hyper-parameter tuning, model validation, edge sampling, computation of edge embeddings and model validation. The framework integrates efficient procedures for edge and non-edge sampling and can be used to easily evaluate any off-the-shelf embedding method. The framework is freely available as a Python toolbox. Finally, demonstrating the usefulness of EvalNE in practice, we conduct an empirical study in which we try to replicate and analyse experimental sections of several influential papers.

## 1 Introduction

Link prediction is an important task with applications in a wide range of fields such as computer science, social sciences, biology, and medicine (Garcia-Gasulla et al., 2015; Lichtenwalter & Chawla, 2012; Lü & Zhou, 2011; Yang et al., 2015). It amounts to estimating the likelihood for the existence of edges, between pairs of nodes that do not form an edge in the input graph. Many Network Embedding (NE) methods (e.g., Belkin & Niyogi, 2002; Cao et al., 2015; Gao et al., 2018; Grover & Leskovec, 2016; Kang et al., 2018; Lai et al., 2017; Ou et al., 2016; Perozzi et al., 2014; Tang et al., 2015) have recently been applied to solving link prediction problems, showing promising results. These methods map nodes in the network to vectors in $\mathbb{R}^d$. This embedding is then used for a variety of tasks such as visualization, multi-label classification, clustering or link prediction.

**The challenges of evaluating NE methods for link prediction** We argue that the practical performance of most NE methods is poorly understood and that experiments in many papers are difficult to compare due to variation in experimental setup and evaluation procedures. In this paper, we focus on a number of difficulties specific to *the evaluation of NE methods for link prediction*. Link prediction is a particularly challenging task to evaluate as it involve a number design choices, which can confound the results and are prone to errors.

*1) Train-test splitting of graphs* For example, a typical implicit assumption is that the input graph is not complete, and the purpose is to accurately predict the missing edges. To evaluate the performance of an NE method for link prediction, one thus needs an (incomplete) training graph along with a (more) complete version of that graph for testing. Much research has been devoted to determining the best approach to generate these training graphs (Garcia-Gasulla et al., 2015; Lichtenwalter & Chawla, 2012; Yang et al., 2015). Strong theoretical and empirical evidence suggest that in order to fairly evaluate link prediction methods, snapshots of the network at different points in time should be used for training and testing. In this way, the link prediction methods are tested on the natural evolutions of the networks. However, the availability of such snapshots is uncommon and raises additional questions, such as how to choose the time intervals for splitting the network.

For these reasons, authors typically resort to sampling sets of edges from the input graphs and using the resulting sub-graphs for training (Gao et al., 2018; Grover & Leskovec, 2016; Kang et al., 2018; Lai et al., 2017). The remaining edges are used as positive test examples. The process of sampling edges is not standardized and varies between scientific works. The relative sizes of the train and test sets, for example, is a user-defined parameter which varies significantly. In Grover & Leskovec (2016); Kang et al. (2018) the authors use a 50-50 train-test split, in (Gao et al., 2018) a 60-40, in Lai et al. (2017) an 80-20 and in Wang et al. (2016) values ranging from 30-70 up to 80-20.

A related problem is that, in addition to the 'positive' train and test edges, often also 'negative' edges (or non-edges) are required. Sometimes these are used to derive the embedding, while in other cases they are used only to train the classifier that predicts links. These sets of non-edges can be selected according to different strategies (Kotnis & Nastase) and can be of various sizes.

*2) From node embeddings to edge predictions* Furthermore, most NE methods simply provide node embeddings. From these, edge embeddings need to be derived prior to performing predictions. There are several approaches (Chen et al., 2018) for deriving edge embeddings which also seem to have a strong impact on the performance of different methods (Grover & Leskovec, 2016).

*3) Evaluation measures* Also the metrics used to evaluate the accuracy varies, e.g., from AUC-ROC (Kang et al., 2018), to precision-recall (Wei et al., 2017), to precision@k (Wang et al., 2016).

*4) Parameter tuning* Finally, it appears to be common practice in recent literature to use recommended default settings for existing methods, while tuning the hyper-parameters for the method being introduced. When the recommended default settings were informed by experiments on other graphs than those used in the study at hand, this can paint an unduly unfavorable picture.

For none of the above problems a gold standard exists which would clear the preferable choice. There are many valid choices and this, together with the fact that not all values of parameters and choices are usually reported have led to a situation where *no one can see the forest for the trees*.

**Contributions** To address this problems, we propose EvalNE, a framework that simplifies the complex and time consuming process of evaluating NE methods for link prediction. EvalNE automates many parts of the evaluation process: hyper-parameter tuning, selection of train and test edges, negative sampling, and more. The framework: (1) Implements guidelines from research in the area of link prediction evaluation and sampling (Garcia-Gasulla et al., 2015; Lichtenwalter & Chawla, 2012; Yang et al., 2015). (2) Includes (novel) efficient edge and non-edge sampling algorithms. (3) Provides the most widely used edge embedding methods. (4) Evaluates the scalability and accuracy of methods, through wall clock time and a range of fixed-threshold metrics and threshold curves. (5) Integrates in a single operation the evaluation of any number of NE methods coded in any language on an unbounded array of networks. (6) Finally, EvalNE ensures reproducibility and comparability of the evaluations and results. *By making it easier to reproduce, compare, and understand evaluation pipelines, we regain the ability to assess the real strengths and weaknesses of existing NE methods, paving the way for more rapid and more reliable progress in the area.*

The remainder of this paper is organized as follows. In Section 2 we discuss related work. Section 3 presents the proposed evaluation framework including the novel edge samplig strategy. Empirical results of attempts to reproduce evaluation setups of several papers and of the proposed edge sampling strategy are reported in Section 4. Finally Section 5 concludes this paper. The open-source toolbox is freely available at [scrapped for anonymity].

## 2 RELATED WORK

The evaluation of link prediction has been studied in several works (Garcia-Gasulla et al., 2015; Lichtenwalter & Chawla, 2012; Martínez et al., 2016; Yang et al., 2015). These papers focus on the impact of different train set sampling strategies, negative sampling, and fair evaluations criteria.

Link prediction as evaluation for the representations learned by a NE algorithm was first introduced in the pioneering work of Grover & Leskovec (2016). The survey by Zhang et al. (2018) points out the importance of link prediction as an application of network representation learning. The authors also signal the inconsistencies in the evaluation of different NE approaches and conduct an empirical analysis of many of these methods on a wide range of datasets. Their empirical study, however, only

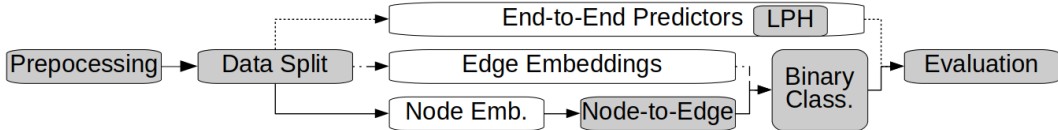

Figure 1: Diagram of the types of methods which can be evaluated using EvalNE. Gray blocks represent modules provided by the library and white blocks are the user-specified methods to be evaluated. The library allows for the evaluation of end-to-end prediction methods (several LP heuristics are included as baselines here), edge embedding methods, and node embedding methods.

focuses on vertex classification and clustering. The importance of standard evaluation frameworks as tools to bridge the gap between research and application is discussed in (Hamilton et al., 2017).

To the best of out knowledge only two frameworks for the evaluation of NE methods currently exist. `OpenNE` is a recently proposed toolbox for evaluating NE methods on multi-label classification. The toolbox also includes implementations of several state-of-the-art embedding methods. `GEM` (Goyal & Ferrara, 2018) is a similar framework which also implements a variety of embedding methods and includes basic evaluation on multi-label classification, visualization, and link prediction tasks. These frameworks, however, are focused on the implementations of embedding methods rather than the evaluation pipeline. Furthermore, these libraries are limited to the NE methods provided by the authors or require new implementations which comply with pre-defined interfaces.

## 3 EVALNE

In this section we discuss the key aspects of EvalNE. The framework has been designed as a pipeline of interconnected and interchangeable building blocks, as illustrated by Figure 1. The modular structure of our framework simplifies code maintenance and the addition of new features, and allows for flexible model evaluation. EvalNE can be used to evaluate methods providing node embeddings, edge embeddings, or similarity scores (we include in this category the link prediction heuristics). Next we describe the frameworks and its components in more detail as well as the software design.

### 3.1 EVALUATION FRAMEWORK

The core building blocks of EvalNE are the data split and model evaluation. These bocks constitute the most basic pipeline for assessing the quality of link predictions. However, in order to extend the evaluation process to other types of embedding methods we also provide building blocks for data manipulation and preprocessing, learning edge embeddings from node embeddings, binary classification and a range of LP heuristics which can be used as baselines.

Before presenting in detail each of these building blocks we introduce some needed notation. We represent an undirected weighted network as $G = (V, E, W)$ with vertex set $V = \{1, \ldots, N\}$, edge set $E \subseteq V \times V$ and weight matrix $W \in \mathbb{R}^{N \times N}$. Edges are represented as unordered pairs $e = (u, v) \in E$ with weights $w_e \in [0, \infty)$. $E_{train}$ and $E_{test}$ denote the training and testing edge sets. We represent a $d$-dimensional node embedding as $\mathbf{X} = (x_1, x_2, \ldots, x_N)$ where $\mathbf{X} \in \mathbb{R}^{N \times d}$.

**Preprocessing** The toolbox offers a variety of functions to load, store, and manipulate networks. These include methods to prune nodes based on degree, remove self-loops, relabel nodes, obtain sets of specific types of edges, restrict networks to their main connected components and obtain common network statistics. The preprocessing functions of EvalNE build on top of and can be used in combination with those provided by other mainstream software packages (e.g. Networkx).

**Data split** As pointed out in Section 1, in order to perform link prediction on a given input graph $G$, sets of train and test edges are required. The set of training edges is generally required to span all nodes in $G$ and induce a train graph $G_{train}$ with a single connected component, because embeddings of independent components will be far away from and unrelated to each other.

Most studies resort to a naive algorithm for deriving a connected $G_{train}$. The procedure removes edges from an input graph iteratively until the required number of train edges remain. The removed

edges are used as test samples. In each iteration, the connectedness of the graph is checked and an edge is only removed if it does not cause the graph to become disconnected. This requirement is generally satisfied by running a Breadth First Search (BFS) on the graph after each edge removal, which is a costly operation ($\mathcal{O}(|V| + |E|)$).

Integrated in our evaluation framework, we include a novel algorithm to perform the train-test splits which, as we will show in Section 4, is orders of magnitude faster yet equally simple. Our algorithm, also accounts for the fact that the training graph $G_{train}$ must span all nodes in $G$ and contain a single connected component. Given as input an undirected graph $G = (V, E, W)$ and a target number of train edges $m$, the proposed algorithm proceeds as follows:

1. Obtain a uniform spanning tree $ST$ of $G$
2. Initialize the set of training edges $E_{train}$ to all edges in $ST$
3. Add $m - |E_{train}|$ edges to $E_{train}$ selected uniformly at random wihout replacement from $E \setminus E_{train}$

We select a spanning tree uniformly at random from the set of all possible ones using Broder's algorithm (Broder, 1989):

1. Select a random vertex s of $G$ and start a random walk on the graph until every vertex is visited. For each vertex $i \in V \setminus s$ collect the edge $e = (j, i)$ that corresponds to the first entrance to vertex i. Let T be this collection of edges.
2. Output the set T.

On expectation, the complexity of the uniform spanning tree generation is $\mathcal{O}(n \log n)$ (Broder, 1989) and the addition of random edges can be efficiently done in $\mathcal{O}(|E_{train}|)$.

For directed graphs $G^* = (V, E^*, W)$ we first construct an equivalent undirected version $G$ by adding reciprocals for every edge in $E^*$. We then run the same algorithm described above on $G$ and include in the training set only those edges present in the initial directed graph $G^*$. This method results in a weakly connected train graph spanning the same nodes as the original $G^*$.

In addition to train and test edges, sets of train and test *non-edges* (also referred to as *negative samples*) are required in order to evaluate link prediction. These are edges between pairs of vertices $u, v$ such that $e = (u, v) \notin E$. The proposed toolbox can compute these non-edges according to either the *open world* or the *closed world* assumption. The two strategies only differ in the selection of the train non-edges. Under the open world assumption, train non-edges are selected such that they are not in $E_{train}$. Thus, this strategy allows overlapping between train non-edges and test real edges. Under the closed world assumption, we consider the non-edges to be known *a priori* and therefore select them so that they are neither in $E_{train}$ nor in $E_{test}$.

The number of train and test edges and non-edges are user-defined parameters. For the train set size, fractions between 50% and 90% of total edges $E$ of $G$ are recommended. For values below 50%, the resulting train graph will often not preserve the properties of $G$ (Leskovec & Faloutsos, 2006).

**Link prediction heuristics** The LP heuristics take as input a train graph $G_{train} = (V, E_{train}, W)$ spanning the same set of vertices as the initial graph $G$ but containing only the edges in the train set and output scores of node similarity which can be directly used for link prediction. EvalNE includes the following heuristics: common neighbours (CN), Jaccard coefficient (JC), Adamic-Adar index (AA), resource allocation index (RA), preferential attachment (PA) and Katz. In addition to these, a random prediction model is provided for reference. This method simply outputs a uniform random value in $[0, 1]$ as the likelihood of a link between any pair of given vertices $(u, v)$.

For the case of undirected graphs we use the usual definitions of the heuristics (see Appendix A [1]) and for directed graphs we restrict our analysis to either the in-neighbourhood ($\Gamma_i(u)$) or the out-neighbourhood ($\Gamma_o(u)$) of the nodes $u \in V$.

**Node to edge embedding** Unlike for the abovementioned LP heuristics where the output can be directly used for link prediction, for NE methods this is generally not the case. Most authors only

---

[1] https://www.dropbox.com/sh/8whq0di1sb9pz8m/AAD11DIrWRWOjtwuVum_TxnGa?dl=0

release code to compute node embeddings. Thus, an additional step of learning edge embeddings is required in order to perform link predictions via binary classification.

The edge representations can be learned in an unsupervised fashion, for any given edge $e = (u, v)$, by applying a binary operator $\circ$ over the embeddings of nodes $u$ and $v$, i.e. $x_u$ and $x_v$ respectively: $x_{(u,v)} = x_u \circ x_v$. Grover & Leskovec (2016) propose the following alternatives for the operator $\circ$ which we include in our evaluation framework: average, hadamard, weighted $L_1$ and weighted $L_2$ (See Appendix B for equations). Additional user-defined operators can be also easily integrated.

**Binary classification** Most NE methods (e.g., Gao et al., 2018; Grover & Leskovec, 2016; Kang et al., 2018; Lai et al., 2017) rely on a logistic regression classifier to predict the probability of links given the edge embeddings. In EvalNE we implement logistic regression with 10-fold cross validation. The framework, however, is flexible and allows for any other binary classifier to be used.

**Evaluation** The proposed framework can evaluate the scalability, parameter sensitivity and accuracy of embedding methods. We asses the scalability directly by measuring wall clock time. The performance can be easily reported for different values of embedding dimensionality and hyperparameter values. Finally, the link prediction accuracy is reported using two types of metrics: fixed-threshold metrics and threshold curves. Fixed-threshold metrics summarize method performance to single values. EvalNE provides the following: confusion matrix (TP, FN, FP, TN), precision, recall, fallout, miss, accuracy, F-score and AUC-ROC. Threshold curves present the performance of the methods for a range of threshold values between 0 and 1. EvalNE includes precision-recall (Lichtenwalter & Chawla, 2012) and receiver operating curves (Fawcett, 2004). The framework provides recommendations of the most suitable metrics based on the evaluation setup.

### 3.2 SOFTWARE DESIGN

The EvalNE framework is provided as a Python toolbox compatible with Python2 and Python3 which can run on Linux, MacOS, and Microsoft Windows. The toolbox depends only on a small number of popular open-source Python packages, and the coding style and documentation comply with strict formats. The documentation provided contains instructions on the installation, examples of high-level and low-level use and integration with existing code.

EvalNE can be used both as a command line tool and an API. As a command line tool it exposes a configuration file which determines the complete experimental setup a user wants to test, including methods to evaluate, networks and edge splits. For convenience, several pre-filled configuration files that reproduce experimental sections of influential NE papers are provided. When used as an API, the framework exposes a modular design with blocks providing independent and self contained functionalities. The user interacts with the library through an evaluator object that integrates all the building blocks and orchestrates the method evaluation pipeline.

## 4 EXPERIMENTS

This section aims to demonstrate the usefulness and flexibility of EvalNE. To this end we have selected and replicated the experimental sections of four papers from the NE literature, i.e. Node2vec (Grover & Leskovec, 2016), CNE (Kang et al., 2018), PRUNE (Lai et al., 2017) and SDNE (Wang et al., 2016). We also report experiments comparing the proposed edge sampling algorithm to the naive approach.

**Experimental setups** The experimental settings we aimed to replicate are as follows:

*Node2vec* This work used the following embedding methods and link prediction heuristics in the evaluation: Node2vec, DeepWalk, LINE, Spectral Clustering, CN, JC, AA, and PA. Experiments are performed on the Facebook, PPI, and AstroPh datasets with 50-50 train-test splits. The same number of edges and non-edges are used throughout the experiment and the edge embedding methods used are average, hadamard, weighted $L_1$ and weighted $L_1$. Results are reported in terms of AUC-ROC.

*PRUNE* The experiments include Node2vec, DeepWalk, LINE, SDNE, PRUNE and NRCL. The networks used are Webspam, HepPH and FB-wallpost and the train-test fraction is 0.8 with similar

| Dataset | Category | $|V|$ | $|E|$ |
|---|---|---|---|
| Facebook | Social | 4039 | 88234 |
| FB-wallpost | Social | 43953 | 262631 |
| BlogCatalog | Social | 10312 | 333983 |
| StudentDB | Social | 395 | 3423 |
| HepPh | Citation | 34546 | 421578 |
| AstroPh | Collaboration | 17903 | 196972 |
| GR-QC | Collaboration | 4158 | 26844 |
| PPI | Biological | 3852 | 37841 |
| Wikipedia | Language | 4777 | 92295 |

Table 1: All the datasets used for evaluation.

|  | Facebook | PPI | arXiv |
|---|---|---|---|
| CN | 0.0226 | 0.0031 | 0.0189 |
| JC | 0.0134 | 0.0066 | 0.0178 |
| AA | 0.0245 | 0.0031 | 0.0189 |
| PA | 0.0242 | 0.0321 | 0.0393 |

Table 2: Absolute difference in recall between results obtained using the naive and proposed edge split.

|  |  | Facebook | PPI | arXiv |
|---|---|---|---|---|
|  | CN | 0.81 (+0.14) | 0.71 (+0.06) | 0.82 (+0.13) |
|  | JC | 0.88 (+0.04) | 0.70 (+0.04) | 0.81 (+0.12) |
|  | AA | 0.83 (+0.13) | 0.71 (+0.06) | 0.83 (+0.12) |
|  | PA | 0.71 (+0.04) | 0.67 (+0.13) | 0.70 (+0.08) |
| Avg. | DeepWalk | 0.72 (−0.01) | 0.69 (+0.08) | 0.71 (+0.01) |
|  | LINE | 0.70 (−0.03) | 0.63 (+0.12) | 0.65 (+0.14) |
|  | node2vec | 0.73 (−0.01) | 0.75 (−0.01) | 0.72 (−0.02) |
| Had. | DeepWalk | 0.97 (−0.03) | **0.74 (−0.2)** | 0.93 (−0.12) |
|  | LINE | 0.95 (−0.06) | 0.72 (−0.01) | 0.89 (+0.05) |
|  | node2vec | 0.97 (0.0) | **0.77 (−0.17)** | 0.94 (−0.05) |
| W $L_1$ | DeepWalk | 0.96 (−0.01) | 0.60 (+0.14) | 0.83 (+0.09) |
|  | LINE | **0.95 (−0.33)** | 0.70 (−0.01) | **0.88 (−0.28)** |
|  | node2vec | 0.96 (−0.01) | 0.63 (−0.03) | 0.85 (+0.03) |
| W $L_2$ | DeepWalk | 0.96 (−0.01) | 0.61 (+0.14) | 0.83 (+0.09) |
|  | LINE | **0.95 (−0.33)** | 0.71 (−0.02) | **0.89 (−0.27)** |
|  | node2vec | 0.96 (0.0) | 0.62 (−0.02) | 0.85 (+0.03) |

Table 3: Reported AUC-ROC values in the node2vec paper and difference to our reproduced values.

sizes for the non-edge sets. The node to edge embedding operator used is not reported by the authors and the results are presented in terms of AUC-ROC. In this setting, the network used are directed.

*CNE* The evaluation setup is like the Node2vec paper, with the following differences: CNE and Metapath2vec are also evaluated; and evaluation is done also on the BlogCatalog, Wikipedia, and StudentDB networks. Results are reported for the hadamard operator only. This paper comes from a research group that we have occasional collaborations with. We had access to the full code for their evaluation pipeline, but the EvalNE codebase was implemented completely separately.

*SDNE* The paper reports experiments for the following NE methods: SDNE, LINE, DeepWalk, GraRep, LapEig and CN. The link prediction experiments are performed on the GR-QC dataset with a 0.85 train-test fraction. The results are reported only for the hadamard operator and in terms of precision@k for a colection of values between 2 and 10000. The number of train non-edges is the same as that of train edges while *all* the remaining non-endges in the graph are used for testing.

We replicated these setting with the following exceptions (1) for node2vec we did not include spectral clustering and (2) for PRUNE we could not obtain NRCL and lacked the computational resources to evaluate the webspam dataset. In all cases we report the same metrics as the original papers and, except were reported differently, average the results over three repetitions. We use the closed-world assumption for the non-edges, logistic regression for binary classification and tuned the hyper-parameters equivalently as reported in each paper. All experiments were run using specific configuration files created for each setting which will be made public.

Regarding the implementations, we evaluated the LP heuristics included in EvalNE; original code by the authors for Deepwalk, Node2vec, LINE, PRUNE, CNE, and Metapath2vec, and for the remaining ones, the implementations in the `OpenNE`[2] library (See Appendix C for references to exact implementations). Table 1 contains the main characteristics of all the networks used. Network sizes are those corresponding to the main connected components.

---

[2] https://github.com/thunlp/OpenNE

|  | DeepWalk | LINE | Node2vec | SDNE | PRUNE |
|---|---|---|---|---|---|
| Hep-Ph | 0.80 (−0.05) | 0.80 (−0.09) | 0.81 (−0.05) | 0.75 (−0.04) | 0.86 (−0.03) |
| FB-wallpost | 0.83 (+0.01) | 0.78 (+0.09) | 0.85 (−0.09) | 0.86 (−0.02) | 0.88 (−0.01) |

Table 4: Reported AUC-ROC values in the PRUNE paper and difference to our reproduced values.

|  | Facebook | PPI | arXiv | BlogCatalog | wikipedia | studentdb |
|---|---|---|---|---|---|---|
| CN | 0.97 (+0.01) | 0.77 (0.0) | 0.94 (+0.01) | 0.92 (+0.01) | 0.84 (0.0) | 0.42 (−0.01) |
| JS | 0.97 (+0.01) | 0.76 (0.0) | 0.94 (+0.01) | 0.78 (0.0) | 0.50 (−0.01) | 0.42 (−0.01) |
| AA | 0.98 (0.0) | 0.77 (+0.01) | 0.94 (+0.01) | 0.93 (0.0) | 0.86 (+0.01) | 0.42 (−0.01) |
| PA | 0.83 (+0.01) | 0.89 (+0.01) | 0.86 (+0.01) | 0.95 (0.0) | 0.91 (+0.01) | 0.91 (+0.01) |
| DeepWalk | 0.98 (0.0) | 0.64 (+0.01) | 0.92 (+0.01) | 0.61 (−0.01) | 0.56 (0.0) | 0.76 (+0.03) |
| LINE | 0.95 (0.0) | 0.75 (+0.01) | 0.98 (0.0) | 0.76 (+0.01) | 0.71 (0.0) | 0.86 (−0.02) |
| Node2vec | 0.99 (0.0) | 0.68 (+0.02) | 0.97 (0.0) | 0.73 (−0.05) | 0.67 (−0.08) | 0.83 (0.0) |
| Metapath | **0.74 (+0.17)** | 0.85 (0.0) | 0.83 (+0.02) | 0.91 (0.0) | 0.83 (+0.02) | 0.92 (−0.01) |
| CNE(unif.) | 0.99 (0.0) | 0.89 (+0.01) | 0.99 (0.0) | 0.92 (+0.01) | 0.84 (0.0) | 0.93 (0.0) |
| CNE(deg.) | 0.99 (0.0) | 0.91 (0.0) | 0.99 (0.0) | 0.96 (0.0) | 0.92 (−0.01) | 0.94 (0.0) |

Table 5: Reported AUC-ROC values in the CNE paper and difference to our reproduced values.

## 4.1 EVALUATION RESULTS

For each of the experimental settings reproduced we present a table containing the original values reported in the paper, and between parentheses the difference between our result and this value (Tables 3–6). Positive values in parentheses thus indicate that the our results are higher than the ones reported in the original papers by that margin, while negative values indicate the opposite.

The results obtained show that our experiments align fairly well with those reported in the CNE and PRUNE papers. The only exception is the result of Metapath2vec on the Facebook dataset, which substantially differs from the CNE paper. For Node2vec and SDNE, the differences are larger and occasionally severe (differences over 0.15 are marked in bold in all tables). Possible explanation for this are the use of different implementations of the NE methods, different not reported default parameters or the use of parallelization. We have studied the effect of each of these possible causes and found 1) important differences in accuracy when comparing different implementations of LINE and Node2vec on identical settings, 2) different initial default parameters for distinct implmentations of deepwalk, and 3) performance degradation for Metapath2vec when using parallelization (See Appendix D for more details).

In addition, we have also observed important differences when computing the AUC-ROC directly from class labels or from class probabilities. In order to reproduce the CNE experiments we used class probabilities (which is the appropriate choice), while for Node2vec class labels appear to have been used. These results illustrate the need to: (a) create reproducible pipelines for experiments, and (b) report specifics about the parameter settings and precise implementations used in the evaluation.

## 4.2 EDGE SAMPLING EVALUATION

We now present our results comparing the proposed edge and non-edge sampling strategy in terms of accuracy and scalability to the naive approach. For the accuracy experiment we selected the Facebook, PPI, and arXiv datasets. We used both the proposed and naive edge split strategies and performed link predictions with the CN, JC, AA, and PA heuristics. We output all the metrics available in EvalNE and compute their absolute difference. Recall is the metric presenting the highest deviation so we collect these results in Table 2. For the AUC-ROC, one of the most widely used metrics, the maximum deviation is of 0.01 reached on the arXiv dataset by the PA heuristic. Consistently, across methods and datasets, the results obtained with the naive approach were slightly higher than those using our method.

Regarding the scalability experiments, in Figure 2 we summarize the execution time in seconds of both methods on four different datasets. This experiment shows that the proposed method is several order of magnitude faster than the naive approach and independent on the number of train and test edges required. The execution time of the naive approach further increases as more test edges are required. In Figure 3 we select the BlogCatalog dataset and restrict it to sub-graphs of different

| | prec@100 | prec@200 | prec@300 | prec@500 | prec@800 | prec@1000 | prec@10000 |
|---|---|---|---|---|---|---|---|
| SDNE | 1 $(-0.00)$ | 1 $(-0.00)$ | 1 $(+0.01)$ | 0.99 $(-0.00)$ | 0.97 $(+0.03)$ | 0.91 $(+0.09)$ | 0.25 $(-0.12)$ |
| LINE | 1 $(-0.00)$ | 1 $(-0.00)$ | 0.99 $(+0.01)$ | 0.93 $(+0.06)$ | 0.74 $(+0.26)$ | 0.79 $(+0.11)$ | 0.21 $(-0.13)$ |
| DeepWalk | **0.6 $(+0.40)$** | **0.55 $(+0.45)$** | **0.44 $(+0.56)$** | **0.34 $(+0.65)$** | **0.29 $(+0.70)$** | **0.29 $(+0.71)$** | 0.15 $(+0.01)$ |
| GraRep | **0.04 $(+0.96)$** | **0.03 $(+0.97)$** | **0.03 $(+0.97)$** | **0.04 $(+0.96)$** | **0.03 $(+0.97)$** | **0.03 $(+0.97)$** | **0.19 $(+0.16)$** |
| CN | 1 $(-0.00)$ | 0.96 $(+0.03)$ | 0.96 $(+0.03)$ | 0.98 $(-0.00)$ | 0.87 $(+0.05)$ | 0.79 $(+0.09)$ | 0.19 $(-0.00)$ |
| LapEig | 0.93 $(+0.07)$ | 0.85 $(+0.15)$ | **0.82 $(+0.17)$** | **0.66 $(+0.34)$** | **0.46 $(+0.53)$** | **0.39 $(+0.48)$** | 0.05 $(+0.15)$ |

Table 6: Reported precision@k values in the SDNE paper and difference to our reproduced values.

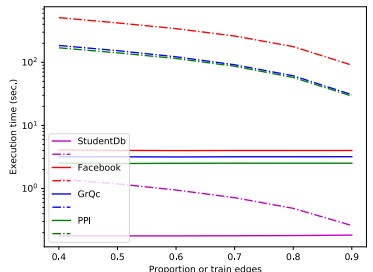

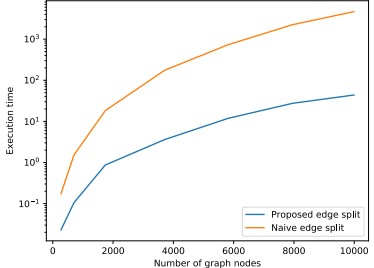

Figure 2: Execution times of proposed (solid lines) and naive (dashed lines) edge split methods w.r.t. the proportion of train edges.

Figure 3: Execution times of proposed and naive edge sampling methods on sub-graphs of BlogCatalog of increasing sizes.

number of nodes ranging from $400$ up to $10000$. The execution times using both sampling strategies are reported in Figure 3.

For our proposed edge sampling we also evaluated the variance in method performance for different number of experiment repetitions (or equivalently, the effect of averaging the results over different number of edge splits). In this case we used the same datasets, NE methods and LP heuristics as in the Node2vec experiment and 50-50 and 90-10 train-test splits. We compared the results obtained in a single run with those averaged over 5 independent runs for both split fractions. For the 50-50 train-test split the average difference observed over all methods, datasets and metrics is $3.4 \cdot 10^{-3}$ with a variance of $1.8 \cdot 10^{-5}$ and a maximum difference of $0.0293$. For the 90-10 split, the average difference is $3.0 \cdot 10^{-3}$ the variance of $1.0 \cdot 10^{-5}$ and maximum difference $0.0186$. These results indicate that a single train and test split already gives a sufficiently good estimate of the generalization error of the models evaluated. Thus, experiment repeats are not needed for networks of similar sizes. These observations seem to hold for different train-test split fractions.

## 5    CONCLUSIONS

The recent surge of research in the area of network embeddings has resulted in a wide variety of data sets, metrics, and setups for evaluating and comparing the utility of embedding methods. Comparability across studies is lacking and not all evaluations are equally sound. This highlights the need for specific tools and pipelines to ensure the correct evaluation of these methods. Particularly, the use of representation learning for link prediction tasks requires train and test sampling, non-edge sampling, and in many cases selection of edge embedding methods and binary classifiers. The evaluation procedure, thus, becomes an ensemble of tasks which allow for many errors or inconsistencies.

In this work we have proposed EvalNE, a novel framework that can be used to evaluate any network embedding method for link prediction. Our pipeline automates the selection of train and test edge sets, simplifies the process of tuning model parameters and reports the accuracy of the methods according to many criteria. Our experiments highlight the importance of the edge sampling strategy and parameter tuning for evaluating NE methods. We have also introduced a scalable procedure to select edge sets from given networks and showed empirically that is orders or magnitude faster than the naive approaches used in recent literature.

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
