# OpenReview forum: "EvalNE: A Framework for Evaluating Network Embeddings on Link Prediction"
_ICLR.cc/2019/Workshop/RML — RML 2019_

### Official Review · AnonReviewer1 · 2019-04-01
**Interesting evaluation**

**Rating:** 3
**Confidence:** 3

**Review:**

The paper presents EvalNE, a framework to evaluate the performance of node embedding methods on link prediction tasks, The framework abstracts away many challenges that naturally arise when evaluating node embedding approaches. For instance, the framework supports various techniques to sample nodes for training the node embedding model. It also comes with different baselines-  both learning-based and heuristics-based, thus encouraging reproducibility of prior work on new datasets.

One caveat is that there is no "one-size fits all" solution. For example, I can think of cases where we would want the graph to be sparse (or maybe even disconnected). Nonetheless, the work is a useful contribution for standardizing evaluation of node embedding techniques.

There are minor grammatical errors like "this errors", "samplig", "out knowledge" etc

---

### Decision · Program_Chairs · 2019-04-05
**Acceptance Decision**

Accept